# Embracing AI in academia: A mixed methods study of nursing students' and educators' perspectives on using ChatGPT

**Ebtsam Aly Abou Hashish**[1,2,3¤a☺]*, **Sharifah Abdulmuttalib Alsayed**[1,2¤a‡],
**Noura Mohamed Fadl Abdel Razek**[3¤b☺]

**1** College of Nursing - Jeddah, King Saud bin Abdul-Aziz University for Health Sciences, Jeddah, Saudi Arabia, **2** King Abdullah International Medical Research Center, Jeddah, Saudi Arabia, **3** Faculty of Nursing, Alexandria University, Alexandria, Egypt

☺ These authors contributed equally to this work.
‡ This author also contributed equally to this work.
¤aCurrent Address: College of Nursing – Jeddah, King Saud bin Abdul-Aziz University for Health Sciences, Jeddah, Saudi Arabia
¤bCurrent Address: Faculty of Nursing, Alexandria University, Alexandria, Egypt
* ebtsam_ss@hotmail.com

## Abstract

### Background

The integration of artificial intelligence (AI) tools such as ChatGPT is reshaping academic practice, particularly in nursing education. Understanding how nursing students and educators perceive and interact with ChatGPT is essential for its responsible and effective use in both academic and clinical contexts. This study aimed to explore knowledge, perceptions, attitudes, and concerns related to ChatGPT among nursing students and educators and to identify potential factors associated with its use in academia.

### Methods

A convergent parallel mixed-methods design was conducted at a Saudi nursing college. Quantitative data were collected from a convenience sample of 240 students and 40 nurse educators using validated self-reported questionnaires. Data were analyzed using descriptive statistics, ANOVA, Pearson's correlation, and regression analysis. Qualitative data were gathered through semi-structured interviews with 20 students and 15 educators and analyzed thematically.

### Results

Participants demonstrated moderate knowledge and generally positive attitudes and perceptions toward ChatGPT. Educators expressed stronger ethical concerns, particularly regarding plagiarism, over-reliance, and data accuracy. Regression analysis

**Data availability statement:** All relevant data are within the manuscript and its supporting information.

**Funding:** The author(s) received no specific funding for this work.

**Competing interests:** The authors have declared that no competing interests exist.

demonstrated that knowledge significantly predicted perceptions and attitudes, with strong predictive power (p < 0.001). Also, ChatGPT experience, academic level, and years of experience as significant predictors of knowledge, perceptions, and attitudes (p ≤ 0.05), with ChatGPT experience showing the strongest predictive power. Thematic analysis yielded four main themes and 22 subthemes: uses of ChatGPT, benefits, concerns, and suggestions for improvement.

## Conclusion

ChatGPT holds promising potential in nursing education, supporting academic productivity and digital competence. However, concerns about ethical use, content accuracy, and discipline alignment remain. Integrating AI literacy training, ethical guidelines, and discipline-specific adaptations is essential to maximize ChatGPT's benefits and support its safe and effective use in nursing academia.

## Introduction

Digital transformation refers to changes driven by the adoption of digital technologies across all areas of an organization [1]. It involves the integration of digital tools, such as Artificial Intelligence (AI), to enhance operations, efficiency, and scalability [2]. One of the most notable AI advancements is ChatGPT, which has emerged as a transformative force in various sectors, including academia. ChatGPT revolutionizes processes by improving teaching, research, and learning efficiencies [3]. In nursing academia, AI tools like ChatGPT are reshaping educational practices by providing support for writing, research, and student engagement [4]. Recent years have seen significant investment in AI technologies, with institutions exploring the potential of tools like ChatGPT to enhance educational outcomes [5].

OpenAI's ChatGPT, a state-of-the-art conversational AI model, is one AI tool that is gaining attention. Powered by deep learning algorithms and trained on vast datasets, ChatGPT can generate human-like responses to text inputs [6]. The integration of ChatGPT into academic settings presents numerous opportunities for innovation and advancement, particularly in disciplines like nursing education [7]. Nursing education involves comprehensive training in critical thinking, clinical decision-making, and evidence-based practice [8]. ChatGPT has the potential to augment these educational endeavors by offering instant access to vast repositories of knowledge, facilitating interactive learning experiences, and supporting research endeavors [9].

In nursing education, integrating AI holds promise for addressing challenges such as information overload, limited access to resources, and the need for personalized learning experiences [10]. The advent of ChatGPT brings forth numerous benefits and opportunities for students and educators. It holds promise in aiding students through learning facilitation, enhancing digital literacy, and fostering critical thinking regarding AI's role in healthcare integration [11]. ChatGPT also allows for more personalized learning experiences by analyzing a student's performance and providing tailored feedback, helping educators identify areas where students may be struggling

and adjust their teaching strategies accordingly [12,13]. Additionally, ChatGPT can simulate patient scenarios, enabling nursing students to practice their skills in a safe and controlled environment [14].

Likewise, nurse educators have the opportunity to integrate ChatGPT into their curriculum and utilize it for formative or summative assessments [15]. Nurse educators can incorporate ChatGPT into classroom discussions, assignments, and assessments. They can use ChatGPT to facilitate interactive learning experiences, provide personalized feedback, and offer on-demand access to resources. By creatively integrating ChatGPT into various aspects of nursing education, educators can enhance the learning experience, promote critical thinking, and prepare students for the technology-driven healthcare environment [16]. This technology not only enhances the educational experience but also supports advancements in nursing research, ultimately contributing to better healthcare outcomes. In nursing research, ChatGPT's ability to analyze vast amounts of data provides researchers with new insights into patient care, disease prevention, and treatment options [12,13]. Therefore, it is crucial for nurse educators to prioritize faculty development to gain a thorough understanding of AI technologies and effectively incorporate them into teaching practices [17].

While ChatGPT offers significant advantages, there are also limitations and challenges associated with its use. For instance, while it aids in learning, concerns about its accuracy and reliability persist. Additionally, both educators and students may become overly reliant on the tool, potentially impeding the development of independent critical thinking skills. These challenges, while concerning, do not diminish the significant potential ChatGPT holds to support educational endeavors [13,18]. The successful implementation of AI in nursing education requires careful consideration of multiple factors, including the perspectives of educators and students, the technical infrastructure, and ethical considerations [17]. These elements must be continuously examined through ongoing research to ensure effective and responsible integration of ChatGPT. Understanding the perceptions, attitudes, and concerns of both students and educators is essential for making informed decisions and formulating effective implementation strategies.

Moreover, existing literature largely focuses on the technical aspects of AI adoption in academia, often neglecting the human and socio-cultural factors that play a crucial role in its acceptance and effectiveness [19]. Addressing these gaps is crucial for the future of AI integration in nursing education.

## Theoretical underpinning and conceptual framework

This study is guided by the Technology Acceptance Model (TAM) (Davis, 1989) [20] which provides a theoretical foundation for understanding how individuals adopt and use technology. TAM posits that Perceived Usefulness and Perceived Ease of Use are the primary determinants of a user's attitude toward technology, shaping their intention to use and actual adoption behavior. In the context of AI integration in academia, TAM helps to explain how nursing students and educators perceive ChatGPT's role in enhancing learning, research, and academic tasks.

Perceived Usefulness, refers to the degree to which users believe that ChatGPT can improve their academic performance, such as facilitating writing, critical thinking, and research support. Perceived Ease of Use, relates to how effortlessly users expect to interact with ChatGPT, influencing their acceptance or reluctance to engage with AI-powered tools. These constructs provide insights into attitudes, usability concerns, and adoption barriers, particularly within nursing education, where the balance between technological advancement and ethical considerations is crucial [20,21].

Given the increasing integration of ChatGPT in educational settings, TAM serves as a valuable framework for assessing how nursing students and educators perceive its role in learning, research, and academic tasks. The model helps explain why individuals may be more inclined to adopt ChatGPT if they find it beneficial for writing, critical thinking, and research assistance, and if they perceive it as easy to use with minimal cognitive effort [21,22]. Furthermore, the expansion of TAM to include factors such as social influence and ethical concerns aligns with the study's focus on nursing education. Educators and students may evaluate ChatGPT not only based on its usability but also on factors such as trustworthiness, ethical risks (e.g., plagiarism, data privacy), and institutional policies. Prior studies on AI adoption in education have successfully employed TAM to explore students' acceptance of AI-powered learning tools, highlighting the

importance of training, guidelines, and institutional support in fostering ethical and effective AI use [22,23]. Applying TAM in this study allows for a structured examination of attitudes, concerns, and adoption barriers, providing insights into how AI tools like ChatGPT can be responsibly integrated into nursing education while enhancing learning efficiency and maintaining academic integrity.

Building upon this theoretical underpinning, the conceptual framework for this study explores four key dimensions: knowledge, perceptions, attitudes, and concerns, while also identifying factors influencing ChatGPT's use through qualitative analysis. *Knowledge* focuses on participants' understanding of ChatGPT's capabilities, applications, strengths, and limitations, highlighting their familiarity with the tool and its potential to support academic tasks and research [24]. *Perception* examines how participants evaluate ChatGPT's usefulness, reliability, and effectiveness in an academic setting, offering insights into its perceived value as an educational tool [24,25]. *Attitudes* reflect participants' confidence, enthusiasm, and willingness to incorporate ChatGPT into their academic activities, indicating their level of acceptance and readiness for AI adoption [26]. *Concerns* and ethical considerations address worries regarding ChatGPT's accuracy, privacy, bias, and over-reliance, emphasizing the need for transparency and ethical guidelines to ensure responsible use [24,27].

Beyond these quantitative dimensions, the qualitative component of the study explores additional factors shaping ChatGPT's use from both students' and educators' perspectives. This holistic approach offers a comprehensive understanding of ChatGPT's role in academia, ensuring its thoughtful and ethical integration into educational practices. By combining TAM's predictive framework with a broader conceptual model, this study aims to support educators and students in adapting to a rapidly evolving technology-driven academic environment, balancing innovation with academic integrity.

## Significance of the study

Digital literacy is a fundamental competency in nursing informatics, particularly in an era of rapid digital transformation. Nursing education must prioritize digital literacy to prepare students and faculty as both knowledge consumers and innovators [1]. ChatGPT, as an AI-powered tool, offers significant potential to enhance learning by providing quick access to information, facilitating research, and fostering critical thinking [28]. As healthcare increasingly incorporates technology, nurse educators play a pivotal role in ensuring students are equipped to deliver safe, high-quality patient care in a technology-driven environment [29].

Several studies have explored the role of ChatGPT and other AI-powered tools in education, focusing on their impact on learning engagement, academic integrity, and research productivity. Abdulai and Hung [30] discussed the ethical dilemmas AI introduces into nursing education, particularly regarding privacy, plagiarism, and over-reliance on AI-generated content. Castonguay et al. [31] highlighted the transformative potential of ChatGPT in clinical and academic learning, emphasizing how AI can support research, automate tasks, and assist with complex problem-solving. Abou Hashish and Alnajjar [1] examined nursing students' digital health literacy and attitude toward AI and stressed the necessity of preparing future nurses for digital transformation, highlighting both the benefits and challenges of AI integration in healthcare education. Abdelhafiz et al. [25] investigated knowledge, perceptions, and attitude of researchers towards using ChatGPT. Although these studies provide valuable insights into AI adoption, they have predominantly focused on general academic contexts rather than discipline-specific settings like nursing education. Most research on ChatGPT's role in education has examined students' knowledge, perceptions, and attitudes broadly, without considering the unique demands of nursing education, which integrates critical thinking, ethical decision-making, and hands-on clinical experience. While ChatGPT has been recognized for enhancing research efficiency and automating tasks [25,31], concerns persist about its accuracy, ethical risks, and potential for diminishing students' independent learning abilities [30,31].

## Research gap and rationale for the study

In this regard, although AI adoption has been increasingly examined in education, there remains a clear gap in understanding how nursing students and educators specifically perceive and interact with ChatGPT [32]. Unlike learners in general

academic domains, nursing students engage with AI in both theoretical and clinical settings, where decision-making, patient safety, and ethical practice are essential. In addition, there is limited exploration of how educators—who play a key role in implementing, guiding, and monitoring AI tools integration in curricula—view its use, benefits, and potential risks [33,34].

This study responds to these gaps by offering a nursing-specific analysis of ChatGPT adoption. It goes beyond general perspectives by including both students and educators voices and exploring how ChatGPT is used and perceived in both academic and clinical contexts. Through a mixed-methods approach, the study explores the educational advantages of ChatGPT, the ethical concerns such as plagiarism, privacy, and over-reliance, and the institutional mechanisms needed to guide its responsible use. The findings aim to inform evidence-based strategies for integrating AI into nursing education while ensuring alignment with professional values and standards, offering insights that support digital transformation in healthcare education.

### Aim of the Study

This study aimed to explore nursing students' and educators' perspectives on using ChatGPT in academia through a mixed-methods approach. The specific objectives of the study were to explore nursing students' and educators' knowledge, perceptions, attitudes, and concerns regarding ChatGPT and to identify potential benefits, challenges, ethical considerations, and recommendations associated with its use in academia.

## Methods

### Research setting and design

This study was conducted at the College of Nursing—Jeddah (CONJ), affiliated with King Saud bin Abdul-Aziz University for Health Sciences (KSAU-HS) in Saudi Arabia. CONJ admits female nursing students and offers a four-year nursing program followed by an internship year. A convergent parallel mixed-methods design was employed, integrating quantitative and qualitative data collected simultaneously [35]. The quantitative component utilized a cross-sectional design with self-reported questionnaires to measure perceptions of ChatGPT among nursing students and educators. In parallel, the qualitative component employed a descriptive exploratory design, using semi-structured interviews to gain deeper insights into the perspectives of nursing students and educators on ChatGPT's use in academia. This convergent parallel design allowed for simultaneous data collection and cross-validation. Quantitative and qualitative data were analyzed independently and then merged during interpretation [36]. This approach provided a comprehensive understanding of ChatGPT's role, benefits, and challenges in nursing education and research. The study followed the STROBE (Strengthening the Reporting of Observational Studies in Epidemiology) guidelines for the quantitative component and the COREQ (Consolidated Criteria for Reporting Qualitative Research) guidelines for the qualitative component, particularly for reporting semi-structured interviews.

### Sampling and participants

**Quantitative Component.** The target population included all undergraduate nursing students and academic nurse educators at CONJ. See (Fig 1) for participants' recruitment flow chart.

For nursing students, eligibility criteria included being 18 years or older and enrolled in levels 4–8 during the 2024/2025 academic year. A total of 240 students participated, achieving an 80% response rate. While all 300 eligible students were invited, a sample size calculation was conducted using the Raosoft Sample Size Calculator to ensure statistical validity. With a 95% confidence interval, a 5% margin of error, and a significance level of 0.05, the recommended minimum sample size was 169 participants. However, to enhance representativeness, reduce sampling bias, and account for potential missing or incomplete data, a larger sample of 240 students was analyzed. This approach ensured broad participation across different academic levels, thereby increasing the generalizability of findings within the nursing education context. Additionally, a pre-test phase involving 20 students was conducted to validate the study instruments.

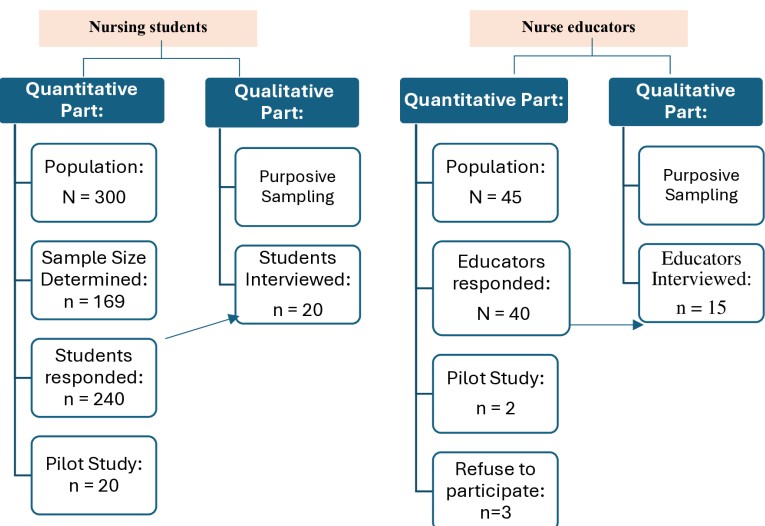

**Fig 1. Participants' recruitment flow chart.**

For nurse educators, the study included all academic staff at CONJ, encompassing professors, associate professors, assistant professors, lecturers, and teaching assistants from diverse cultural backgrounds (e.g., Saudi, Egyptian, Indian, Filipino, and Malaysian). Of the 45 educators invited, 40 completed the survey, yielding a high response rate of 88.9%.

**Qualitative Component.** The qualitative phase involved semi-structured interviews with a purposive subsample of nursing students (n = 20) and educators (n = 15) who voluntarily agreed to participate in the qualitative phase. Purposive sampling was used to capture a range of perspectives across different academic levels and faculty ranks. Nurse educators were chosen based on their enrollment status, experience with ChatGPT, and voluntary participation, ensuring the relevance of the data to the study objectives. The sampling approach was guided by Polkinghorne's recommendation [37] to interview 5–15 individuals with direct experience of the phenomenon under study. Interviews continued until data saturation was reached, defined as the point where no new information emerged [38]. Data collection commenced with a pilot interview and concluded after 35 interviews (20 students and 15 educators), ensuring comprehensive coverage of relevant insights and experiences.

## Research instruments

The study employed a researcher-developed self-reported questionnaire and semi-structured interviews, grounded in the literature review [1,3,7,16–18,25–32] to comprehensively explore participants' perspectives on ChatGPT use in academic settings.

### Tool 1: ChatGPT questionnaire was structured into two parts

*Part 1:* Demographic Information This section aimed to collect basic demographic and academic information from both nursing students and educators. It included questions on age, academic level (for students), academic position (for educators), previous experience with AI tools, and areas of use. *Part 2:* Knowledge, Perceptions, Attitudes, and Concerns Related to ChatGPT. It consisted of four parts, each containing 10 statements. Responses were measured on a 5-point Likert scale ranging from 1 (strongly disagree) to 5 (strongly agree).

## Tool 2: Semi-structured Interview Guide

Semi-structured interviews were used to gain in-depth insights into participants' perspectives on using ChatGPT in academia. This qualitative data collection method combines pre-determined open-ended questions with the flexibility to explore emerging topics during the conversation. Unlike fully structured interviews that strictly follow a set list of questions, semi-structured interviews (SSIs) allow interviewers to probe deeper into issues as they arise, making them especially effective for exploring complex phenomena from participants' viewpoints.

The interview guide was developed by the researchers and consisted of open-ended questions designed to elicit detailed responses about participants' experiences, perceptions, and expectations regarding ChatGPT in academic settings. These questions encouraged participants to share nuanced insights drawn from their personal experiences. The semi-structured format also allowed interviewers to ask follow-up or probing questions based on nursing participants' responses, facilitating a deeper understanding of the topics under discussion. The questions include:

- In what manner do you use ChatGPT for academic nursing tasks?

- In which areas of nursing research or education do you think ChatGPT could be most beneficial? Please provide examples of ChatGPT's effective applications in your field.

- What are the main benefits of using ChatGPT in nursing academia?

- What do you perceive as the main weakness or disadvantage of using ChatGPT in your academic nursing work?

- What are your main concerns about using ChatGPT in nursing research and academia?

- Can you provide examples of how ChatGPT has positively impacted your nursing research or academic tasks?

- What improvements would you suggest for the future use of ChatGPT?

## Validity and reliability

To ensure the validity and reliability of the study instruments, a rigorous process was followed. The instruments were initially validated in their original English format by academic experts, ensuring their alignment with the study objectives and relevance to assessing perceptions of ChatGPT among nursing students and educators. A pre-testing phase involving 20 students and 2 educators identified minor adjustments that improved clarity and usability for the main study.

For the quantitative questionnaire, internal reliability was assessed using Cronbach's alpha, which yielded a value of 0.853, indicating good consistency across items at a statistically significant level ($p \leq 0.05$). While the semi-structured interview guide, designed to capture in-depth perspectives on ChatGPT's use in academia, was validated by qualitative research experts through peer review. This process ensured the guide's questions were clear, relevant, and capable of eliciting detailed and meaningful responses. Moreover, two pilot interviews were conducted to refine the interview guide and assess the interviewers' proficiency, ensuring effective facilitation during the main study. The interviews were conducted by experienced faculty members with expertise in qualitative research and interview techniques, which further enhanced the credibility of the data collection process.

## Data collection

Data collection began after obtaining approval from both the CONJ Research Committee and the King Abdullah International Medical Research Center's (KAIMRC) Institutional Review Board under approval number NRJ24/034/8. Both quantitative and qualitative data were collected using methods that ensured flexibility and participant convenience. For the quantitative component, participants were provided with both paper-based and electronic versions of the questionnaire. Paper-based questionnaires were distributed to students during class sessions and to educators in their offices, with

completed forms collected immediately. The electronic version was shared via email and WhatsApp, allowing a one-week submission window for online completion. To maintain data integrity and prevent duplicate responses, each participant was assigned a unique identifier, and the electronic system tracked submissions. Completing the questionnaire required approximately 20 minutes per participant.

For the qualitative component, semi-structured interviews were conducted either face-to-face or via Microsoft Teams, depending on participants' preferences. Before each interview, participants were briefed on the study's purpose, ethical considerations, confidentiality measures, voluntary participation, and consent requirements. Interviews lasted between 30 and 45 minutes, were audio-recorded with participants' permission, and comprehensive notes were taken immediately afterward. Data collection continued until saturation was reached, defined as the point where no new themes emerged. Interview data analysis began concurrently with data collection after the first interview, with transcripts prepared to ensure accuracy and facilitate the identification of emerging themes. The data collection process was conducted over two months, from October 7 to December 7, 2024, during the first semester of the 2024–2025 academic year.

## Data analysis

Quantitative data were analyzed using the Statistical Package for Social Sciences (SPSS) version 25. Descriptive statistics, including means and standard deviations, were used to summarize the results. Analysis of Variance (ANOVA) was employed to compare group means across demographic categories. Pearson's correlation assessed relationships among variables, and regression analysis tested the predictive capacity of the independent variable (knowledge) on the dependent variables (perceptions, attitudes, and concerns). Multiple regression analysis was also conducted to identify predictors of participants' knowledge, perception, attitude, and concerns toward ChatGPT. The significance level for all statistical analyses was set at $p \leq 0.05$.

*Qualitative Data:* Qualitative data analysis employed thematic analysis following Braun and Clarke's framework [39] to identify recurring themes and patterns in participants' responses. The analysis process began with verbatim transcription of interviews, followed by repeated readings to ensure familiarization with the data. Initial codes were generated by identifying significant phrases and recurring ideas, such as "improved writing skills," "support for research," and "limitations in accuracy." These codes were organized into broader categories and refined through an iterative process.

To enhance trustworthiness and ensure intercoder reliability, two independent researchers manually coded the data separately. After the initial coding phase, the researchers compared their codes and engaged in discussions to resolve any discrepancies. A consensus approach was adopted, where disagreements were addressed through discussion until a mutual agreement was reached on final themes and categories. This process minimized bias and ensured that the thematic structure accurately represented participants' perspectives. Additionally, reflexivity was maintained throughout the analysis by keeping an audit trail of coding decisions, ensuring transparency and rigor in the thematic development process. Thematic analysis was conducted iteratively, with data collection and analysis occurring simultaneously, allowing researchers to refine emerging themes based on new data.

**Integration of quantitative and qualitative data.** The integration of quantitative and qualitative data was achieved through triangulation to enhance validity and provide a more comprehensive understanding of the research findings. Both data sets were analyzed concurrently, and the results were compared and synthesized during the interpretation process. This approach allowed for a deeper exploration of the participants' perspectives on the use of ChatGPT in academia, facilitating the identification of convergences and divergences between the quantitative trends and qualitative insights.

## Ethical considerations

The study was approved by both the CONJ Research Committee and the King Abdullah International Medical Research Center's (KAIMRC) Institutional Review Board under approval number NRJ24/034/8. Prior to participation, students and

educators were fully informed about the study's purpose, including their right to refuse or withdraw at any time without any consequences. Written informed consent was obtained from all participants, and strict measures were implemented to ensure data privacy and confidentiality. Anonymity was maintained by using pseudonyms or participant numbers in the reporting of interview data.

### Rigor and trustworthiness

To ensure the rigor and trustworthiness of the qualitative data, the study adhered to Shenton's [40] four criteria: credibility, transferability, dependability, and confirmability. Credibility was enhanced through member checking, triangulation of interview data with relevant literature, and frequent team reviews to ensure accurate representation of participants' voices. Researchers also established participant selection criteria and provided feedback to the interviewer to refine the process. Transferability was supported by providing thick descriptions of the study context, methodologies, and participants to allow readers to assess applicability to other settings. Dependability was ensured through an audit trail documenting research decision, peer reviews, and consistency checks by a peer researcher. Confirmability was achieved through reflexivity, minimizing bias, and using direct participant quotes to support findings [41].

## Results

### Demographic characteristics

A total of 40 female nurse educators and 240 nursing students successfully participated in this study. Nurse educators ages range from 23 to 60 years, with a mean age of 35.5±7.6 years. Teaching experience varies from 1 to 30 years, with a mean of 10.8±4.2 years. The majority of participants are Saudi (67.5%), while 32.5% are non-Saudi. In terms of academic rank, 60% are assistant professors. Regarding ChatGPT usage, 70% of educators reported moderate use of the tool. For nursing students, the mean age is 20.5±1.2 years, indicating that most students are in their early 20s. Regarding previous experience with ChatGPT, the majority of students have used it in some capacity, with 84 students (35.0%) reporting moderate use and 54 students (22.5%) reporting extensive use. See supplementary file for tables 1 and 2 for more details about demographic characteristics of participants.

### Uses of ChatGPT among nursing students and nurse educators

The primary academic and research activities for which nurse educators use ChatGPT include writing research papers (55.0%), followed by curriculum development, presentation preparation, and student support (32.5%). ChatGPT is also used for creating assignments (27.5%), reflecting its integration into both instructional design and student engagement (Fig 2). While, among nursing students, the leading reasons for using ChatGPT are study assistance (72.5%), assignment preparation (55.0%), and research assistance (45.0%), highlighting its function as an academic support tool throughout their learning journey (Fig 3).

### Knowledge, perception, attitude, and concerns of ChatGPT among nurse educators and students

Table 1 shows that nurse educators reported a slightly higher understanding and knowledge of ChatGPT, with an average score of 3.27 (56.84%) compared to 3.11 (52.67%) for students. In terms of the perception of ChatGPT's utility, educators exhibited a slightly more favorable view, with an average score of 3.65 (66.32%) compared to students' 3.58 (64.56%). Both groups were similarly enthusiastic about the tool, with educators scoring 3.72 (67.92%) and students scoring 3.75 (68.84%). Despite these slight differences, the results were not statistically significant. However, a significant difference was found in concerns and ethical considerations, where educators expressed stronger concerns. Educators scored 3.92 (72.99%), while students scored 3.67 (66.67%), with a statistically significant difference (p=0.003). (Fig 4).

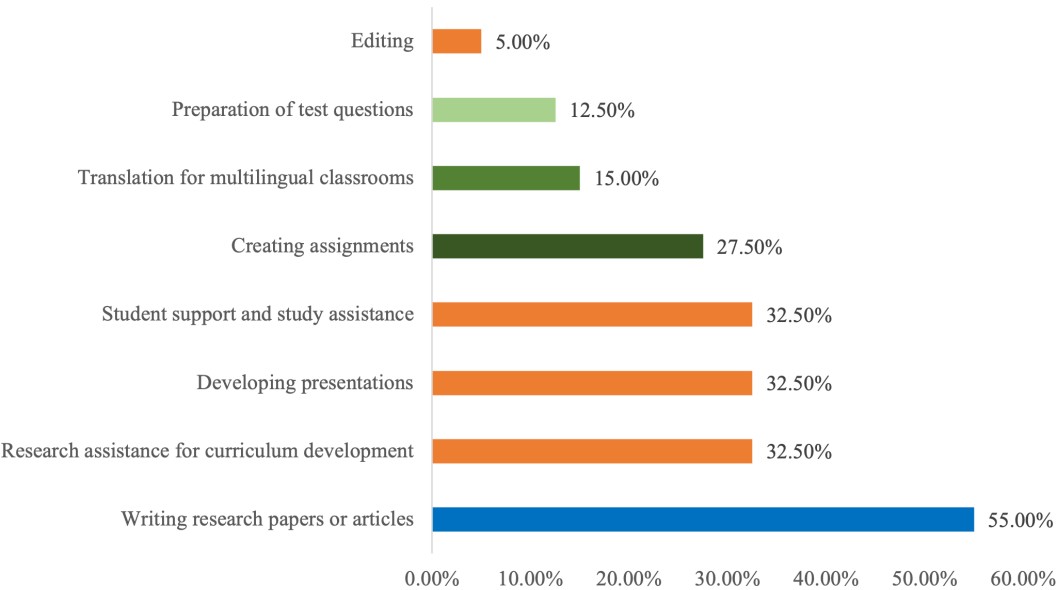

**Fig 2. Primary uses of ChatGPT among nurse educators in nursing academia.**

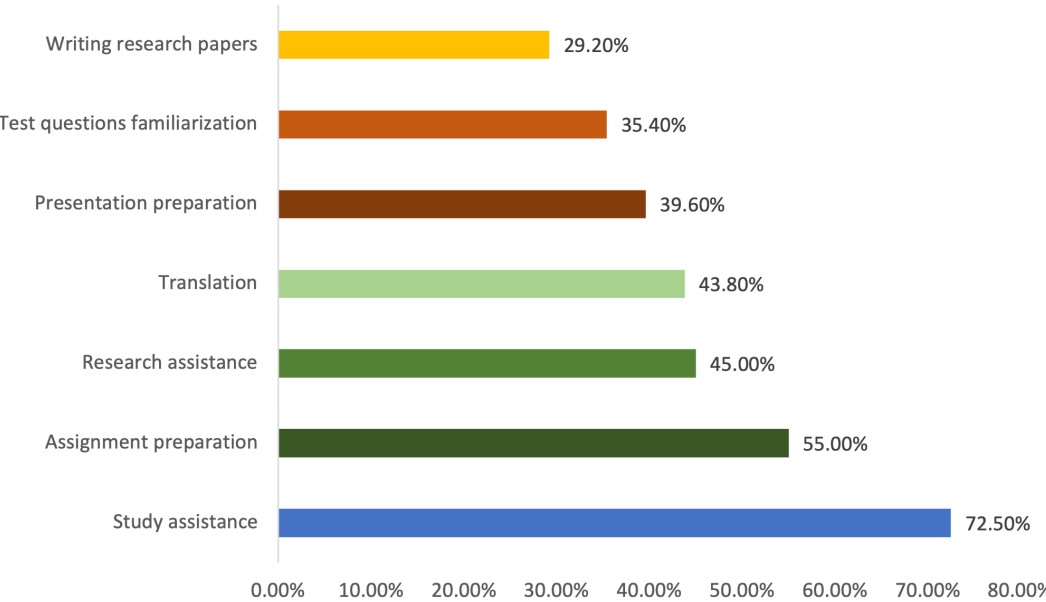

**Fig 3. Primary uses of ChatGPT among nursing students.**

## Regression among the studied variables

Table 2 presents the correlations between knowledge, perception, attitude, and concerns regarding ChatGPT among both educators and students. Among educators, the strongest correlation was observed between knowledge and concerns ($r = 0.825$, $p < 0.001$), indicating that greater knowledge was associated with fewer concerns. Perception was also strongly

**Table 1. Perceived Knowledge, perception, attitude, and concerns of ChatGPT among nurse educators and students.**

| Items | Educators (N=40) | Students (N=240) | t | p |
|---|---|---|---|---|
| Knowledge about ChatGPT | | | | |
| Average Score | 3.27±0.75 | 3.11±0.92 | 1.845 | 0.066 |
| Percent score | 56.84±18.77 | 52.67±22.92 | | |
| Perception of ChatGPT utility | | | | |
| Average Score | 3.65±0.50 | 3.58±0.90 | 0.861 | 0.390 |
| Percent score | 66.32±12.45 | 64.56±22.49 | | |
| Attitude toward using ChatGPT | | | | |
| Average Score | 3.72±0.65 | 3.75±1.0 | 0.399 | 0.690 |
| Percent score | 67.92±16.25 | 68.84±24.90 | | |
| Concerns and ethical considerations | | | | |
| Average Score | 3.92±0.64 | 3.67±0.89 | 2.969* | 0.003* |
| Percent score | 72.99±15.97 | 66.67±22.34 | | |

Notes: t: Student t-test*: Statistically significant at p ≤ 0.05

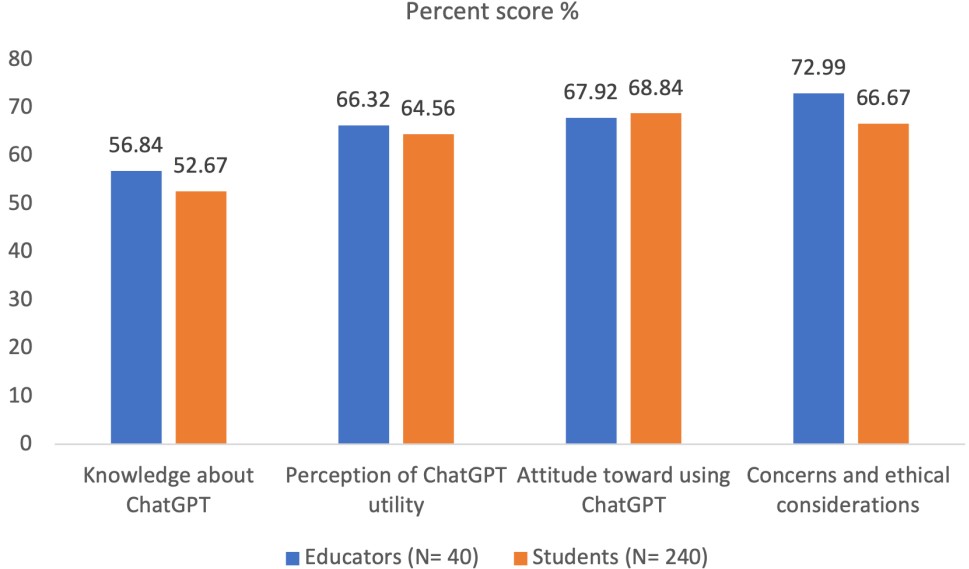

**Fig 4. Perceived knowledge, perception, attitude, and concerns of ChatGPT among participants.**

correlated with attitude (r=0.711, p<0.001). The correlation between knowledge and attitude was weaker (r=0.207, p=0.013), though still statistically significant. Correlations between other variables were moderate but significant, including perception and concerns (r=0.298, p<0.001), and attitude and concerns (r=0.240, p=0.004). Among students, all correlations were strong and statistically significant at p<0.001. The highest correlation was between perception and attitude (r=0.916), followed by knowledge and attitude (r=0.894), knowledge and perception (r=0.851), and attitude and concerns (r=0.578). The weakest but still significant correlation was between knowledge and concerns (r=0.369, p<0.001), and perception and concerns (r=0.461, p<0.001).

**Table 2. Correlation among the studied variables in educators and students' sample.**

| Variables | Educators | | | | Students | | | |
|---|---|---|---|---|---|---|---|---|
| | Knowledge | Perception | Attitude | Concerns | Knowledge | Perception | Attitude | Concerns |
| **Knowledge**<br>r<br>p<br>R² | – | 0.341<br>< 0.001*<br>0.12 | 0.207<br>0.013*<br>0.04 | 0.825<br>< 0.001*<br>0.68 | – | 0.851<br>< 0.001*<br>0.72 | 0.894<br>0.001*<br>0.80 | 0.369<br>0.001*<br>0.14 |
| **Perception**<br>r<br>p<br>R² | 0.341<br>< 0.001*<br>0.12 | – | 0.711<br>< 0.001*<br>0.80 | 0.298<br>< 0.001*<br>0.09 | 0.851<br>< 0.001*<br>0.80 | – | 0.916<br>< 0.001*<br>0.84 | 0.461<br><0.001*<br>0.21 |
| **Attitude**<br>r<br>p<br>R² | 0.207<br>0.013*<br>0.04 | 0.711<br>< 0.001*<br>0.80 | – | 0.240<br>0.004*<br>0.06 | 0.894<br>< 0.001*<br>0.80 | 0.916<br><0.001*<br>0.84 | – | 0.578<br>< 0.001*<br>0.14 |
| **Concerns**<br>r<br>p<br>R² | 0.825<br><0.001*<br>0.68 | 0.298<br><0.001*<br>0.09 | 0.240<br>0.004*<br>0.06 | – | 0.369<br>< 0.001*<br>0.14 | 0.461<br><0.001*<br>0.21 | 0.578<br>< 0.001*<br>0.14 | – |

Notes: r: Pearson coefficient*: Statistically significant at $p \leq 0.05$

In addition, Table 3 of the regression analysis revealed that knowledge of ChatGPT significantly predicts perception, attitude, and concerns for both educators and students, with varying strengths of influence. For educators, knowledge had the strongest predictive power for concerns (B = 0.82, p < 0.001), indicating that as educators' knowledge about ChatGPT increases, so do their concerns about its use. Knowledge also positively influenced perception (B = 0.34, p < 0.001) and attitude (B = 0.20, p = 0.013), suggesting that educators with more knowledge tend to have a more positive view and attitude toward using ChatGPT. For students, knowledge strongly predicted attitude (B = 0.89, p < 0.001) and perception (B = 0.85, p < 0.001), highlighting that more knowledgeable students are more likely to have positive perceptions and attitudes toward ChatGPT. Although knowledge also predicted concerns (B = 0.37, p < 0.001) in students, the effect was more moderate compared to educators.

## Participants' characteristics and perceived ChatGPT

Table 4 presents the demographic predictors of nurse educators' knowledge, perception, attitude, and concerns regarding ChatGPT. All listed variables (age, nationality, position, years of experience, and ChatGPT experience) were statistically significant predictors (p ≤ 0.05) across most outcomes. ChatGPT experience had the highest beta values (B = 0.34 to 0.37)

**Table 3. Regression analysis of knowledge as the predictor for perception, attitude, and concerns.**

| Dependent Variable | B | SE | β | t-value | p-value |
|---|---|---|---|---|---|
| Perception (Educators) | 0.34 | 0.10 | 0.34 | 3.40 | < 0.001 |
| Attitude (Educators) | 0.20 | 0.08 | 0.21 | 2.50 | 0.013 |
| Concerns (Educators) | 0.82 | 0.07 | 0.83 | 11.71 | < 0.001 |
| Perception (Students) | 0.85 | 0.05 | 0.85 | 17.00 | < 0.001 |
| Attitude (Students) | 0.89 | 0.04 | 0.89 | 22.25 | < 0.001 |
| Concerns (Students) | 0.37 | 0.10 | 0.37 | 3.70 | < 0.001 |

*β (Standardized Coefficient), B (Coefficient): Unstandardized Regression coefficients,*

*SE: Standard Error P significant at ≤ 0.05*

**Table 4. Regression Analysis of Nurse Educators' variables Predicting ChatGPT-related variables.**

| Predictor Variable | Knowledge | | | Perception | | | Attitude | | | Concerns | | |
|---|---|---|---|---|---|---|---|---|---|---|---|---|
| | B | p | f² | B | p | f² | B | p | f² | B | p | f² |
| Age(<31–40) | 0.32 | 0.03* | 0.06 | 0.28 | 0.05* | 0.05 | 0.35 | 0.03* | 0.07 | 0.30 | 0.04* | 0.06 |
| Nationality (Saudi) | 0.22 | 0.04* | 0.04 | 0.18 | 0.06 | 0.03 | 0.25 | 0.03* | 0.05 | 0.20 | 0.05* | 0.04 |
| Position Assistant/ associate Professor) | 0.35 | 0.04* | 0.07 | 0.32 | 0.04* | 0.06 | 0.34 | 0.04* | 0.07 | 0.33 | 0.04* | 0.07 |
| Years of Exp 11–15 years) | 0.30 | 0.04* | 0.06 | 0.33 | 0.04* | 0.07 | 0.29 | 0.05* | 0.06 | 0.31 | 0.04* | 0.06 |
| ChatGPT (Extensive) | 0.36 | 0.04* | 0.07 | 0.37 | 0.04* | 0.08 | 0.34 | 0.04* | 0.07 | 0.35 | 0.04* | 0.07 |

B: Unstandardized regression coefficient *p ≤ 0.05 is statistically significant, f²: Cohen's effect size for regression; 0.02 = small, 0.15 = medium, 0.35 = large.

and the largest effect sizes (f² = 0.07 to 0.08), indicating that prior exposure to the tool was the strongest predictor of all four outcomes. Position as Assistant and Associate Professors and years of experience (11–15) also showed moderate and consistent effects across domains (B = 0.30–0.35, f² = 0.06–0.07).Nationality and age showed smaller but significant effects. Nationality was a significant predictor for knowledge, attitude, and concerns, with Saudi educators generally scoring slightly higher. Age differences showed that educators aged 31–40 tend to report higher levels of knowledge, perception, and attitude, while those in the 51–60 range had higher concerns, though the effect sizes were modest (f² = 0.05–0.06).

Moreover, Table 5 presents the predictors of nursing students' knowledge, perception, attitude, and concerns regarding ChatGPT. All included variables (age, year of study, and ChatGPT experience) showed statistically significant associations with the outcomes (p ≤ 0.05). ChatGPT experience was the strongest predictor across all domains. Students with moderate experience showed the highest standardized beta values (B = 0.34 to 0.36) and the largest effect sizes (f² = 0.07). Year of study also had a meaningful impact. Fourth-year students scored significantly higher than those in earlier years across all outcomes (B = 0.31–0.33, f² = 0.06–0.07). Also, Age was a moderate but significant predictor. Students aged 21 reported higher knowledge, perception, and attitude (B = 0.28–0.32, f² = 0.05–0.06) compared to other age groups, while those older than 21 expressed slightly greater concerns.

## Qualitative findings

**Thematic Analysis Framework.** The thematic analysis of the qualitative data resulted in the identification of four key themes, each with corresponding subthemes, reflecting both educators' and students' perspectives on the use of ChatGPT in academia. These themes cover uses, benefits, weaknesses and concerns, suggested improvements, and the future of AI. The analysis highlights both shared experiences and distinctive views between the two groups (Table 6 outlines these themes). For a more detailed list of representative quotations, please refer to supplementary file 1 for

**Table 5. Regression Analysis of nursing students' demographic variables Predicting ChatGPT-related variables.**

| Predictor Variable | Knowledge | | | Perception | | | Attitude | | | Concerns | | |
|---|---|---|---|---|---|---|---|---|---|---|---|---|
| | B | p | f² | B | p | f² | B | p | f² | B | p | f² |
| Age (21 vs others) | 0.30 | 0.04* | 0.05 | 0.28 | 0.05* | 0.05 | 0.32 | 0.03* | 0.06 | 0.29 | 0.04* | 0.05 |
| Year of Study (4th) | 0.31 | 0.03* | 0.06 | 0.33 | 0.03* | 0.07 | 0.32 | 0.04* | 0.06 | 0.30 | 0.03* | 0.06 |
| ChatGPT Experience (moderate) | 0.36 | 0.03* | 0.07 | 0.34 | 0.03* | 0.07 | 0.35 | 0.03* | 0.07 | 0.34 | 0.03* | 0.07 |

B: Unstandardized regression coefficient *p ≤ 0.05 is statistically significant, f²: Cohen's effect size for regression; 0.02 = small, 0.15 = medium, 0.35 = large.

**Table 6. Integrated Findings of ChatGPT in Academia—All Themes and Subthemes.**

| Themes (4) | Subthemes (22) | Shared Points (NE & NS) | Points of Difference (NE vs NS) |
|---|---|---|---|
| I.Uses of ChatGPT | 1. Research Writing and Support<br>2. Academic Support<br>3. Teaching and Learning<br>4. Other Applications | Both use ChatGPT for writing tasks, summarizing, and simplifying content. | NE: Focus on teaching ideas; NS: Focus on learning assignments. |
| II.Benefits and Positive Impact | 5. Writing Improvement<br>6. Collaboration and Support<br>7. Time Saving and Efficiency<br>8. Efficient Research Search<br>9. Simplifying Complex Concepts | Both agree on ChatGPT improving writing, saving time, and simplifying complex concepts. | NE: Emphasize research efficiency; NS: Highlight exam preparation. |
| III.Weaknesses and Concerns | 10. Accuracy and Reliability of Information<br>11. Inaccessibility of References<br>12. Lack of Ethical Guidelines<br>13. Plagiarism and Ethical Issues<br>14. Over-reliance and Loss of Skills<br>15. Impact on Research Integrity<br>16. Ethical Concerns in Data Security | Both raise concerns about accuracy, over-reliance, and ethical issues like plagiarism. | NE: Worry about student dependency; NS: Concerned about incomplete or incorrect responses. |
| IV.Suggested Improvements | 17. Valid Referencing, Source Transparency, and Accuracy<br>18. Clear Ethical Guidelines and Responsible Usage<br>19. Specialized and In-Depth Information<br>20. Training, Workshops, and Orientation<br>21. Academic Tools and Features<br>22. Ease of Use and Efficiency | Both suggest adding referencing, ethical guidelines, training, and specialized content. | NE: Stress the need for tools for teaching; NS: Prefer tools for learning and studying. |

table 3. Participant statements are identified using the letter 'PE' for educators and 'PS' for students, followed by a corresponding number (e.g., PE#1 for educators and PS#1 for students).

**Theme I: Uses of ChatGPT**

This theme focuses on the specific ways nurse educators and students employ ChatGPT for academic and professional tasks. It captures the practical applications of ChatGPT as described by both groups across four subthemes: research writing and support, academic support, teaching and learning, and other applications.

I.1.    Research writing and support:

Nurse educators reflect on using ChatGPT for refining research ideas and improving the structure of their academic writing. One educator noted, *"I use ChatGPT for writing research papers to refine ideas and improve grammar."* Similarly, nursing students emphasize using ChatGPT for generating and organizing ideas for research, as stated by one student: *"I use ChatGPT to help me with writing tasks, especially for generating research ideas."*

I.2.    Academic support:

ChatGPT is used to enhance clarity in writing and simplify academic concepts. Nurse educators highlight its role in improving sentence rephrasing, with one stating, *"I use it for sentence rephrasing, especially when I need to improve the clarity of my writing."* Nursing students use it to summarize research topics and break down challenging concepts, as shared by one student: *"ChatGPT helps me find reliable sources and summarize research topics for my assignments."*

I.3.   Teaching and learning:

Educators describe using ChatGPT to generate teaching ideas and make complex medical information more accessible. One educator remarked, *"ChatGPT is a great tool for generating teaching ideas and simplifying complex medical information for my students."* Nursing students, on the other hand, use ChatGPT to break down complex topics into simpler steps, with one reflecting, *"It helps me break down complex topics, especially in nursing techniques, into simpler steps."*

I.4.   Other applications:

Both groups leverage ChatGPT for tasks beyond academics. Educators use it for writing emails and exploring ideas for assignments, as one shared, *"I use ChatGPT for writing emails, translations, and sometimes to search for new ideas related to student assignments."* Students, meanwhile, use ChatGPT for quick information gathering, with one student noting, *"It helps me gather information quickly and efficiently, sometimes also for refining my email and searching for funny ideas."*

**Theme II: Benefits and positive impact of ChatGPT in Academia**

This theme emphasizes the outcomes and advantages reported by nurse educators and students from their use of ChatGPT. It captures the positive impact of ChatGPT as described by both groups across five subthemes: writing improvement and summarizing research, collaboration and support, time saving and efficiency, efficient research search, and simplifying complex concepts.

II.1.   Writing improvement and summarizing research:

ChatGPT's positive impact on writing quality is emphasized by both groups. Nurse educators reflect on how ChatGPT has helped refine grammar and clarity, with one sharing, *"I used ChatGPT to improve the grammar and clarity of my writing, making my academic papers more polished."* Nursing students express similar outcomes, as one stated, *"ChatGPT helps me improve my writing, fix grammar mistakes, and make my work clearer."*

II.2.   Collaboration and support:

ChatGPT is described as a collaborative tool that enhances the refinement of ideas. One nurse educator reflected, *"It feels like having a colleague to bounce ideas off of,"* while a nursing student shared, *"ChatGPT provides useful input to refine my assignments."*

II.3.   Time-saving and efficiency:

Both educators and students highlight how ChatGPT saves time by summarizing content and providing quick answers. An educator stated, *"ChatGPT helps me save time by providing quick answers and summarizing long texts,"* while a student noted, *"ChatGPT saves me time by summarizing things quickly, so I can focus on understanding instead of searching."*

II.4.   Efficient research search:

Nurse educators and students reflect on how ChatGPT streamlines the research process. An educator shared, *"ChatGPT helped me find related research studies much faster than traditional searches,"* and a student remarked, *"I use ChatGPT to identify articles I would have otherwise missed."*

II.5.   Simplifying complex concepts:

ChatGPT's ability to simplify challenging topics is repeatedly noted. Educators emphasize how it makes teaching easier, with one stating, *"I used ChatGPT to simplify medical information into a more understandable format for different audiences."* Nursing students share a similar experience, as one reflected, *"ChatGPT helps me understand complex nursing concepts by breaking them down into simpler terms."*

**Theme III: Weaknesses and concerns about ChatGPT in Academia**

This theme explores the limitations and challenges of using ChatGPT in academic contexts, based on the perspectives of nurse educators and students. The theme is organized into two subthemes: Weaknesses, focusing on functional limitations, and Concerns, emphasizing broader ethical and academic implications.

III. 1.     Weaknesses:

This subtheme highlights specific functional challenges related to ChatGPT's performance and usability.

III.1.1     Accuracy and reliability:

Both educators and students note that ChatGPT does not always provide accurate or reliable information. An educator reflected, *"I've noticed that ChatGPT doesn't always provide the most accurate info, especially when it comes to research or literature,"* while a student stated, *"Sometimes the answers from ChatGPT are incomplete or just flat-out wrong, which makes me double-check everything."*

III.1.2 Inaccessibility of references:

Educators and students express frustration over the reliability and accessibility of references generated by ChatGPT. One educator mentioned, *"The references ChatGPT provides aren't always reliable or accessible, so I can't always verify the information,"* and a student shared, *"I often find that ChatGPT doesn't provide proper sources for the information it gives me."*

III.1.3     Lack of ethical guidelines:

Nurse educators emphasize the need for clear guidelines on appropriate ChatGPT usage. One educator remarked, *"There's a real gap in terms of guidelines on how much we should rely on ChatGPT. We need to know what's okay and what's not."* Students also expressed this sentiment, with one noting, *"It's important to know the limits—when should we use ChatGPT, and when is it better to go with traditional methods?"*

III. 2     Concerns:

This subtheme focuses on broader risks, including ethical challenges and the potential impact on academic integrity.

III. 2.1     Plagiarism and ethical issues:

Both groups highlight concerns about plagiarism and ethical risks associated with ChatGPT-generated content. An educator noted, *"ChatGPT pulls from existing content, and I worry about unintentional copying,"* while a student reflected, *"ChatGPT might cause students to plagiarize if they don't verify sources properly."*

III. 2.2     Over-reliance on ChatGPT:

Over-reliance on ChatGPT is seen as a threat to developing critical thinking and research skills. One educator expressed, *"I worry that if students rely too much on ChatGPT, they'll lose their ability to think critically and do their own research,"* while a student added, *"Using ChatGPT too often could make students lazy and stop them from doing their own thinking or research."*

III. 2.3     Impact on research integrity:

Educators and students fear that ChatGPT may undermine research integrity by encouraging shortcuts and reducing original thought. An educator stated, *"Using ChatGPT could hurt the integrity of research, especially if students start depending on AI-generated content instead of doing their own work,"* while a student shared, *"ChatGPT might encourage lazy habits, where students copy information without thinking critically or checking sources."*

III. 2.4 Ethical concerns in data security:

Both groups raise concerns about data privacy when using ChatGPT. An educator noted, *"There's concern about privacy when using ChatGPT, especially when it comes to sensitive data. It could expose information we don't want to share,"* and a student remarked, *"ChatGPT might expose our academic data if we're not careful, and there's a risk of data breaches."*

**Theme IV: Suggested Improvements for the Future of ChatGPT in Academia**

This theme captures the recommendations from nurse educators and students for enhancing ChatGPT's effectiveness in academic settings. It highlights six subthemes, consolidating overlapping points to present clear and actionable suggestions for improvement.

IV.1.    Valid referencing, source transparency, and accuracy

Educators and students emphasize the need for ChatGPT to provide accurate information and transparent references to enhance academic credibility. An educator noted, *"I would really appreciate it if ChatGPT could provide clear references and mention where the information comes from, so I can be sure it's reliable."* Similarly, a student shared, *"I think it would help if ChatGPT could link to scientific journals or websites to make the research more credible."* Both groups also suggest incorporating plagiarism detection to ensure originality, as reflected by one educator: *"I think it would be really useful if ChatGPT could check for plagiarism or tell us if it's pulling content directly from somewhere else."*

IV.2.    Clear ethical guidelines and responsible usage

The need for clear ethical guidelines is a recurring theme, with both groups highlighting the importance of using ChatGPT responsibly. An educator remarked, *"It's crucial that ChatGPT has clear rules for when it should and shouldn't be used, especially to maintain academic integrity."* Students echoed this concern, with one stating, *"It would be helpful to have some guidelines on how to use ChatGPT ethically, so we don't accidentally misuse it."*

IV.3.    Specialized and in-depth information

Both educators and students desire more tailored content specific to their fields, particularly nursing. An educator shared, *"It would be really helpful if ChatGPT could provide more specialized content that's deeper and more relevant to specific nursing fields."* A student reflected, *"I wish ChatGPT could give more detailed and specific information related to nursing, especially for my research projects."*

IV.4.    Training, workshops, and orientation

Training and workshops to help users understand ChatGPT's capabilities and limitations are seen as essential. One educator suggested, *"I think there should be workshops or training sessions so that we can get the most out of ChatGPT while understanding its limitations."* A student added, *"It would be great if ChatGPT had tutorials or workshops that teach us how to use it effectively for our studies."*

IV.5.    Academic tools and features

Both groups suggest incorporating academic tools, such as flashcards and summaries, to support learning. An educator proposed, *"Having features like academic flashcards to help students review key concepts quickly would be really useful."* Similarly, a student noted, *"It would be helpful if ChatGPT could generate flashcards or summaries to help me revise for exams."*

IV.6.    Ease of use and efficiency

There is a shared call for improving ChatGPT's usability and time-saving capabilities. An educator commented, *"ChatGPT needs to be more intuitive and save time, especially when it comes to organizing resources or structuring my lectures."* A student added, *"It would be awesome if ChatGPT was easier to use, helping me save time when I'm looking for answers or doing my assignments."*

**Integrated Findings: ChatGPT in Academia—All Themes, Subthemes, and Shared/Different Points**

Table 6 provides a comprehensive overview of ChatGPT's role in academia, highlighting its uses, benefits, weaknesses, concerns, and suggested improvements. It captures shared perspectives between nurse educators (NE) and nursing students (NS), while also outlining key differences based on their distinct roles and needs. Both groups agree on ChatGPT's utility in improving writing, saving time, and simplifying complex content, reflecting its broad applicability across academic tasks. However, nurse educators focus more on teaching-related uses and research efficiency, whereas students emphasize its support in assignments, study preparation, and learning key concepts.

The table also sheds light on shared concerns, such as accuracy, plagiarism risks, ethical challenges, and over-reliance on the tool. Divergent perspectives arise as educators worry about students losing critical thinking skills due to dependency, while students are more focused on addressing incomplete or incorrect responses that impact their work. Both groups propose improvements, including the need for valid referencing, ethical guidelines, training, and specialized features. Educators advocate for tools tailored to teaching, while students suggest study aids like flashcards and revision tools. Likewise, (Fig 5) encapsulates the multifaceted aspects of ChatGPT usage in academic settings, structured around

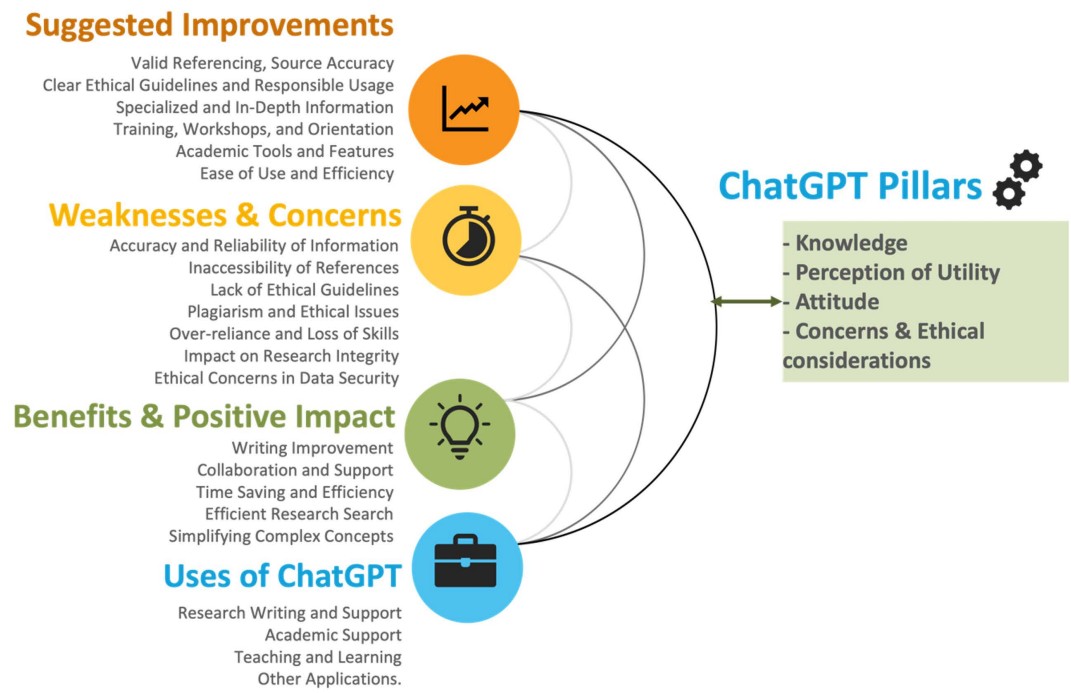

**Fig 5. Summary of determinants, impacts, and recommendations for ChatGPT in Academia.**

four core pillars measured in the quantitative part: knowledge, perception of utility, attitude, and concerns and ethical considerations. It also integrates the emerged themes from the qualitative part. This figure could be considered as a framework for understanding and enhancing ChatGPT's integration into academia, guiding both current applications and future advancements.

## Discussion

The integrated findings highlight ChatGPT's significant role in nursing academia. While the quantitative findings reveal the perceived knowledge, perceptions, attitudes, and their relationships, the qualitative findings complement this by providing in-depth insights into its benefits, weaknesses, and concerns. Together, these findings emphasize the need for targeted improvements to maximize its potential in supporting educational practices.

### Knowledge, perception, attitude, and concern of ChatGPT

The findings revealed that both educators and students exhibit positive attitudes and strong perceptions regarding ChatGPT's utility in academic settings, alongside moderate knowledge about the tool. Both groups recognize the significant potential of ChatGPT in enhancing academic tasks, particularly in writing improvement, research support, and assignment preparation. These findings align with studies by Alkhaqani [12,13] and Vázquez-Cano et al. [42], which emphasize the growing role of AI tools like ChatGPT in boosting productivity and efficiency in both teaching and learning. Both groups expressed enthusiasm for integrating ChatGPT into academia, viewing it as a valuable resource to streamline tasks and improve academic outcomes.

However, the study also revealed that educators expressed heightened concerns about the ethical implications of ChatGPT's use compared to students. These concerns include plagiarism, academic integrity, and the potential erosion of critical thinking. Educators, who are responsible for upholding academic standards, are more attuned to these risks and emphasize the importance of fostering responsible use. Both groups agreed on the need for clear ethical guidelines to mitigate these risks and ensure that ChatGPT enhances academic practices rather than diminishing their quality. These results are backed by detailed feedback from the third theme, which looks at what participants think about the problems and worries related to ChatGPT, and this will be discussed later.

This observation is consistent with previous studies that highlight the ethical challenges posed by AI tools in academia. Farhi et al. [43] and Hammad [44] emphasized concerns about academic dishonesty and its impact on critical thinking, while Salvagno [45] highlighted the broader implications for academic integrity and intellectual rigor. These findings together show that ChatGPT can greatly improve academic work but also brings up ethical issues that need careful rules and plans to handle.

In the same context, the correlation and regression analysis highlight the significant role of knowledge in shaping and predicting attitudes, perceptions of utility, and concerns regarding ChatGPT. Both correlations and regression models show that a higher level of knowledge about ChatGPT is positively associated with more favorable perceptions and attitudes toward its use. This suggests that as users' understanding of the tool increases, so do their concerns about ethical implications such as plagiarism, academic integrity, and the potential impact on critical thinking skills. This aligns with the perspective that as users' understanding and knowledge of new tools and technologies expand, their positive perceptions and attitudes toward their use also increase [35,46,47]. This indicates that providing students with more information about ChatGPT can enhance their trust and confidence in using the tool effectively for academic tasks. These findings are consistent with previous studies, such as those by Kooli [27], Vázquez-Cano et al. [42], and Farhi et al. [43], which emphasize the positive impact of increased familiarity with AI tools on users' attitudes and perceptions. These studies further support the idea that both educators and students benefit from a deeper understanding of ChatGPT, which not only enhances their overall experience with the tool but also helps address concerns about its ethical use in academia.

 

## Factors influencing nurse educators' perceptions of ChatGPT

In the discussion of factors influencing nurse educators' perceptions of ChatGPT, the regression analysis identified prior experience with the tool and professional academic roles, particularly those of Assistant and Associate Professors with increased years of experience, as the most significant predictors. These factors enhanced knowledge, perception, and attitude while reducing concerns. This finding aligns with Ali and Murray [48] and Acosta-Enríquez et al. [49] who noted that familiarity with AI applications positively influences users' confidence and efficiency in educational contexts. Similarly, Lin and Chen [50] found that educators' acceptance of AI technology is shaped by their professional maturity and level of engagement, reinforcing the role of experience and responsibility in AI adoption. These insights underscore the importance of targeted faculty development programs to build AI literacy, particularly among less experienced or older educators, to support the effective integration of tools like ChatGPT in academic settings. On the other hand, a systematic review by Kavandi and Jaana [51] concluded that there is no consistent or strong evidence regarding the influence of socio-demographic variables—such as age, gender, or education level—on technology adoption, suggesting that contextual and experiential factors may be more critical drivers of AI use in academia.

Furthermore, among factors influencing nursing students' perceptions of ChatGPT, regression analysis revealed that academic maturity—reflected by older age (21 years), advanced year of study (fourth year), and increased ChatGPT experience—was significantly associated with higher levels of knowledge, perception, and attitude, and with reduced concerns. These findings suggest that students with greater academic exposure and familiarity with AI tools are more confident and competent in their use. Farhi et al. [43] found that senior students exhibited greater understanding and acceptance of AI technologies in academic contexts. Similarly, Sova et al. [52] emphasized that higher academic levels and exposure to digital tools were key predictors of AI adoption in healthcare education. Fošner [53] also noted that student engagement with AI varies across academic stages, with upper-level students using more advanced AI functions, likely due to increased demands of academic tasks and digital literacy.

## Experiences and uses of ChatGPT

The findings reveal that both educators and students use ChatGPT for academic purposes, including writing improvement, research support, and assignment preparation. Educators frequently utilize ChatGPT for refining research papers, developing curricula, and creating presentations, highlighting its potential to streamline academic tasks. Similarly, students rely on ChatGPT for writing and study tasks, such as summarizing research articles and organizing assignments. These insights align with the qualitative analysis, which underscores ChatGPT's diverse applications in the first theme by both nurse educators and students in research writing, academic support, teaching, and other tasks.

These findings are consistent with prior studies, such as Alkhaqani [12,13] and Salvagno et al. [45], which emphasize ChatGPT's growing role in education, particularly in supporting writing and research. Sallam (26) and Archibald and Clark [54], further note ChatGPT's value in assisting with research paper creation, literature searches, and rephrasing tasks in nursing education. Additionally, Alkhaqani [13] observed ChatGPT's ability to help nursing students manage academic demands, offering support in both theoretical learning and clinical experiences. Vázquez-Cano et al. [42] highlighted its impact on nursing students' academic performance, particularly in research and assignment preparation.

## Benefits and positive impact of ChatGPT

The findings highlight the shared benefits of ChatGPT in academia, as reported by both educators and students. Both groups emphasized its ability to improve writing quality, enhance research efficiency, and simplify complex concepts. ChatGPT's capacity to save time by summarizing research and generating content quickly was particularly valued, enabling both educators and students to focus on higher-order tasks like in-depth analysis. These findings align with studies by Abujaber et al. [29] and Mai et al. [55] which underscored the role of AI tools in boosting academic productivity.

Both educators and students recognized ChatGPT's role in making complex topics more accessible, particularly in nursing education. Educators utilized the tool to create teaching materials that simplify difficult concepts, while students relied on it to break down clinical topics into understandable terms. These observations mirror findings by Abdelhafiz [25] and Naamati-Schneider [56] who emphasized the role of AI tools in clarifying challenging subject matter and improving comprehension.

Additionally, both educators and students viewed ChatGPT as a collaborative tool. Educators described it as an additional colleague that helps refine ideas and support research, while students used it to organize assignments and improve their work. This aligns with research by Alkhaqani [12,13] and Salvagno et al. [45] which highlighted ChatGPT's ability to foster collaboration and enhance academic communication. The tool's impact on research efficiency was another key benefit identified by both groups. Educators and students alike noted its usefulness in tasks like essay writing, article preparation, and summarizing research. These observations align with studies by Vázquez-Cano et al. [42] and Haleem et al. [57], which emphasized ChatGPT's role in supporting research and academic performance.

## Weaknesses and concerns regarding ChatGPT in Academia

The third theme highlights critical weaknesses and concerns about ChatGPT in academia, emphasizing functional limitations and ethical challenges. The findings suggest that both educators and students recognize significant issues with the tool's accuracy and reliability, particularly when verifying sources and references. These concerns align with prior research by Kooli [27] and Gunawan et al. [58] which highlight accuracy as a persistent limitation of AI tools and stress the need for users to critically assess outputs.

*Ethical concerns, such as plagiarism and lack of citation transparency,* also emerged as key issues. Educators and students expressed fears about unintentional copying of AI-generated content, which could compromise academic integrity. These findings echo Najafali et al. [59] who identified plagiarism as a major ethical risk in AI-assisted academic work, and Farhi et al. [43] who warned that such tools might undermine critical thinking and originality.

A major concern was the potential for *over-reliance on ChatGPT*, which educators feared could weaken students' critical thinking and research skills. This concern is consistent with Athilingam and He [60] who highlighted the risks of AI tools diminishing independent problem-solving abilities. Similarly, Abdelhafiz et al. [25] noted participants' apprehensions about AI replacing human roles in research-related tasks, raising questions about the long-term impact on academic development.

Lastly, concerns about *data security* were significant, with both groups emphasizing risks related to privacy and information breaches. These concerns align with findings from Scerri and Morin [61], and Hosseini et al. [62] who emphasized the importance of addressing ethical risks to ensure AI tools do not compromise sensitive academic data. In summary, while ChatGPT offers substantial academic benefits, its weaknesses and ethical challenges must be addressed.

## Suggested Improvements for the Future of ChatGPT in Academia

Both educators and students expressed optimism about the expanding role of AI tools like ChatGPT in academia, anticipating that such technologies will become integral to teaching, learning, and research. They highlighted several key areas for improvement to enhance ChatGPT's functionality, accuracy, and ethical compliance.

A major recommendation was improving academic integrity by ensuring valid referencing, transparent source attribution, and enhanced accuracy. Participants emphasized the importance of addressing inaccuracies and ensuring reliable outputs, aligning with findings by Kooli [27] and Gunawan et al. [58] who stress the need for AI tools to provide verifiable and trustworthy information to uphold academic rigor.

Clear ethical guidelines were also seen as critical for ChatGPT's responsible use. The interviewed educators emphasized the importance of establishing rules to prevent misuse and ensure responsible practices among students, reflecting concerns raised by Farhi et al. [43] and Veras et al. [63] regarding the risks of unregulated AI use in education.

Professional development programs, such as those detailed by Athilingam and He [60] and Scerri and Morin [61] highlight the importance of equipping educators with the knowledge to integrate AI responsibly into academia.

In this respect, existing ethical frameworks in nursing education, such as the American Nurses Association (ANA) Code of Ethics [64] and the International Council of Nurses (ICN) Code of Ethics [65] provide foundational guidance on academic integrity, responsible technology use, and ethical decision-making. Applying these principles to AI use in academia can help develop structured ethical guidelines for ChatGPT and similar AI tools. In this regard, Kucukkaya et al. [66] and Liu et al. [67] emphasized the need for AI-specific competency standards to ensure that nursing students and educators can critically assess AI outputs, integrate them into clinical decisions, and navigate ethical challenges, aligning nursing education with evolving technological demands.

To mitigate *plagiarism and citation concerns*, institutions could implement academic integrity policies [68]. This may include requiring students to disclose AI-assisted work, properly attributing AI-generated content, and integrating plagiarism detection tools capable of recognizing AI-generated text [69]. Educators can also frame ChatGPT as a supplementary tool for enhancing critical thinking rather than as a replacement for independent research and writing [43,63].

Additionally, structured artificial intelligence (AI) literacy programs for both students and educators can promote ethical AI use. Workshops on AI literacy, data privacy, and critical evaluation of AI-generated content can equip users with the necessary skills to navigate ChatGPT responsibly. This aligns with recommendations by Abdelhafiz et al. [25] and Athilingam and He [60] who advocate for structured AI education to prevent over-reliance and ensure ethical academic practices. Professional development programs, as highlighted by Scerri and Morin [61] emphasize the importance of equipping educators with the skills to integrate AI effectively into academia.

Furthermore, concerns about *data security* underscore the need for universities to establish clear guidelines on data-sharing policies and AI-generated content limitations. Encouraging the use of institutionally approved AI platforms with strict privacy controls can help protect sensitive academic information. Additionally, reinforcing awareness about data protection laws such as the General Data Protection Regulation (GDPR) and Health Insurance Portability and Accountability Act (HIPAA) can guide ethical AI use in educational and healthcare settings [61].

Another key suggestion was the need for *specialized and discipline-specific AI applications*, particularly in fields such as nursing education. Educators and students emphasized the value of adapting AI tools to meet the unique requirements of different disciplines, aligning with findings by Kooli [27] and Vázquez-Cano et al. [42] who stress the importance of discipline-specific adaptability in educational technologies.

Training and workshops were widely recommended to enhance understanding and ethical application of ChatGPT. Participants highlighted the need for structured guidance to maximize the tool's potential while preventing misuse. This recommendation reflects findings by Athilingam and He [60] and Scerri and Morin [61], who advocate for comprehensive training programs to help educators and students navigate AI tools effectively. Similar programs, as noted by Haleem et al. [57] emphasize the importance of foundational AI knowledge and practical application skills in fostering AI literacy.

Educators and students also suggested integrating advanced academic tools within ChatGPT, such as flashcards, automated summaries, and customizable learning features to support teaching and learning. They emphasized the need for a more intuitive and efficient user interface to streamline workflows, consistent with broader trends in educational technology [25,43]. These recommendations underscore a shared vision for improving ChatGPT's functionality, ethical compliance, and adaptability. Addressing these key areas—including valid referencing, ethical guidelines, tailored content, robust training, and user-focused enhancements—can enable ChatGPT to better meet the needs of educators and students, ensuring its responsible and impactful integration into academia.

## Strengths and limitations of the study

This study employs a mixed-methods approach, integrating quantitative and qualitative data to provide a comprehensive analysis of ChatGPT's role in nursing education. By combining numerical findings with qualitative insights,

it offers a nuanced understanding of how both educators and students perceive and use the tool. The inclusion of perspectives from both groups ensures a holistic view of ChatGPT's impact on teaching, learning, and research. The study's relevance to current trends in educational technology underscores its contribution to understanding the integration of AI tools in nursing education, particularly their potential to support both theoretical and clinical learning.

However, this study has several limitations. First, the relatively small sample size of 40 educators may not fully capture the diversity of experiences and perspectives in nursing education, potentially limiting the generalizability of the findings to other educators. Since the study was conducted in a single institution in Saudi Arabia, the results may be influenced by specific cultural, educational, and technological factors unique to this setting.

Second, the study was conducted at a single institution in Saudi Arabia, which may limit the generalizability of findings to other educational settings. Institutional policies, faculty training, and students' familiarity with AI tools can differ across universities and countries, which may lead to variations in perceptions of ChatGPT's role in academia. Also, while the themes identified provide important insights, similar studies in different cultural or institutional contexts may yield variations in perceptions of ChatGPT due to this contextual limitation, suggesting that the extent to which ChatGPT enhances or challenges academic practices may vary across settings.

Additionally, the reliance on self-reported data in the quantitative portion introduces potential biases, such as social desirability bias, where participants may provide responses they believe are expected rather than reflecting their true experiences, which could lead to an overestimation of positive perceptions or an underreporting of concerns. Future studies incorporating observational methods, experimental methodologies, or usage analytics may help validate self-reported findings and could provide more objective insights into how ChatGPT is actually utilized in academic settings. Furthermore, the study focused exclusively on ChatGPT, which, while widely used, represents only one of many AI tools in academic settings. The findings may not be fully transferable to other AI applications with different functionalities. Each tool serves distinct purposes in education and may elicit different perceptions and experiences from users. Expanding the scope to include multiple AI tools in future research could provide a more comprehensive understanding of AI integration in nursing education and its impact on teaching and learning outcomes.

## Conclusion

This study offers a nuanced and comprehensive perspective on ChatGPT's transformative role in nursing academia. Both nurse educators and students acknowledged its potential to enhance teaching, streamline academic tasks, improve writing, and support research efficiency. The tool was particularly valued for simplifying complex content and promoting academic productivity. Regression analysis revealed that knowledge significantly predicted perception, attitude, and concerns. Moreover, participant characteristics—such as academic role, experience level, year of study, age, and familiarity with ChatGPT—were key predictors of acceptance. These findings emphasize the importance of academic maturity and prior experience with AI tools in shaping confidence and competency in using ChatGPT effectively.

Despite these benefits, the study also identified ethical concerns, including issues related to plagiarism, over-reliance, academic integrity, and data privacy. Educators expressed stronger concerns, reflecting their critical role in safeguarding academic standards. These findings suggest that integrating ChatGPT into education requires a balanced approach that maximizes benefits while mitigating risks. Establishing clear ethical guidelines, tailoring applications to specific disciplines like nursing, improving AI tool accuracy, and fostering AI literacy through targeted training are essential. These actionable recommendations align with prior research and highlight the importance of thoughtful integration. As AI continues to evolve, embracing tools like ChatGPT—while addressing their limitations—can redefine educational practices, foster innovation, and build digital competencies in both educators and students, paving the way for sustainable growth and transformation in academic environments.

## Recommendations and implications of the study

Based on the study's findings, several recommendations are proposed to support the effective and responsible integration of ChatGPT into nursing academia. First, educational institutions should establish clear ethical guidelines for AI use to mitigate risks such as plagiarism, over-reliance, and academic dishonesty. These policies must be communicated clearly and consistently to both educators and students. Second, targeted training programs—such as AI literacy workshops or integration modules in orientation courses—are needed to enhance users' ability to use ChatGPT effectively and ethically. The findings emphasize that prior experience significantly influences knowledge, perception, and attitudes toward the tool, underscoring the need for structured exposure.

Third, enhancements in ChatGPT's accuracy and content reliability are essential. Both educators and students highlighted the need for valid references and discipline-specific outputs, especially in complex fields like nursing. Developers should consider tailoring ChatGPT's capabilities to meet the specialized academic and clinical needs of healthcare education. Usability improvements—such as interface simplification and time-saving features—were also identified as practical enhancements that could improve students' engagement and efficiency.

At the institutional level, these findings highlight the need for proactive leadership in managing AI integration. Institutions must not only support ethical AI use but also align curriculum design with AI developments. For AI developers, the study offers practical feedback to refine ChatGPT's functionality, reliability, and domain-specific responsiveness. Policymakers are also called to act by introducing frameworks that govern ethical AI use in education while protecting academic integrity. Finally, this study opens avenues for further research. Longitudinal and experimental studies are recommended to evaluate ChatGPT's long-term impact on learning outcomes, critical thinking, and academic performance. Such research is crucial to inform evidence-based practices and guide the sustainable integration of AI into higher education.

## Supporting information

**S1 File. Supplementary file of tables 1–3 for additional data.**
(DOCX)

**S2 File. Raw data.**
(XLSX)

## Acknowledgments

We extend our sincere gratitude to all the participants who contributed their time and insights to this study.

## Author contributions

**Conceptualization:** Ebtsam Aly Abou Hashish, Noura Mohamed Fadl Abdel Razek.

**Data curation:** Ebtsam Aly Abou Hashish, Sharifah Abdulmuttalib Alsayed, Noura Mohamed Fadl Abdel Razek.

**Formal analysis:** Ebtsam Aly Abou Hashish, Sharifah Abdulmuttalib Alsayed, Noura Mohamed Fadl Abdel Razek.

**Investigation:** Ebtsam Aly Abou Hashish, Sharifah Abdulmuttalib Alsayed, Noura Mohamed Fadl Abdel Razek.

**Methodology:** Ebtsam Aly Abou Hashish, Sharifah Abdulmuttalib Alsayed, Noura Mohamed Fadl Abdel Razek.

**Project administration:** Ebtsam Aly Abou Hashish, Noura Mohamed Fadl Abdel Razek.

**Supervision:** Ebtsam Aly Abou Hashish.

**Validation:** Ebtsam Aly Abou Hashish, Sharifah Abdulmuttalib Alsayed, Noura Mohamed Fadl Abdel Razek.

**Visualization:** Ebtsam Aly Abou Hashish, Noura Mohamed Fadl Abdel Razek.

**Writing – original draft:** Ebtsam Aly Abou Hashish, Sharifah Abdulmuttalib Alsayed, Noura Mohamed Fadl Abdel Razek.

**Writing – review & editing:** Ebtsam Aly Abou Hashish.

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
