## [Decision Letter · Decision Letter 0]

PONE-D-24-60162Embracing AI in Academia: A mixed methods study of nursing students' and educators' perspectives on using ChatGPTPLOS ONE

Dear Dr. Abou Hashish,

Thank you for submitting your manuscript to PLOS ONE. After careful consideration, we feel that it has merit but does not fully meet PLOS ONE’s publication criteria as it currently stands. Therefore, we invite you to submit a revised version of the manuscript that addresses the points raised during the review process.

We look forward to receiving your revised manuscript.

Kind regards,

Lucija Gosak

Academic Editor

PLOS ONE

2. Please provide captions for Figure 3 in your manuscript.

3. Please include a complete copy of PLOS’ questionnaire on inclusivity in global research in your revised manuscript. Our policy for research in this area aims to improve transparency in the reporting of research performed outside of researchers’ own country or community. The policy applies to researchers who have travelled to a different country to conduct research, research with Indigenous populations or their lands, and research on cultural artefacts. The questionnaire can also be requested at the journal’s discretion for any other submissions, even if these conditions are not met.  Please find more information on the policy and a link to download a blank copy of the questionnaire here: https://journals.plos.org/plosone/s/best-practices-in-research-reporting. Please upload a completed version of your questionnaire as Supporting Information when you resubmit your manuscript.

5. Please remove your figures from within your manuscript file, leaving only the individual TIFF/EPS image files, uploaded separately. These will be automatically included in the reviewers’ PDF.

Reviewers' comments:

Reviewer's Responses to Questions

**Comments to the Author**

1. Is the manuscript technically sound, and do the data support the conclusions?

Reviewer #1: Yes

Reviewer #2: No

2. Has the statistical analysis been performed appropriately and rigorously? 

Reviewer #1: Yes

Reviewer #2: No

3. Have the authors made all data underlying the findings in their manuscript fully available?

Reviewer #1: Yes

Reviewer #2: Yes

4. Is the manuscript presented in an intelligible fashion and written in standard English?

Reviewer #1: Yes

Reviewer #2: Yes

5. Review Comments to the Author

Reviewer #1: This study provides valuable insights into the role of AI in nursing education. Addressing the concerns outlined below more explicitly will enhance the study’s overall impact.

Thematic Analysis Rigor

1.The qualitative analysis mentions Braun and Clarke’s framework but does not specify how intercoder reliability was ensured. Were multiple researchers involved in coding? If so, how were disagreements resolved? Providing more details would enhance the trustworthiness of the findings.

Limitations

2.While the study acknowledges certain limitations, it should discuss how these might impact the findings and interpretations. For instance, the sample includes students and educators from a single institution. How might this affect the generalizability of the results? Consider discussing whether similar findings could be expected in other cultural or educational settings.

Detailed Discussion on Ethical Concerns

3.The ethical concerns regarding ChatGPT’s use (plagiarism, over-reliance, and privacy issues) are well-identified. However, the study would benefit from a deeper discussion on potential solutions. Could existing ethical frameworks in nursing education provide guidance?

Reviewer #2: Review Report

The study aimed to explore nursing students' and educators' perspectives on using ChatGPT in academia through a mixed-methods approach. However, several concerns raise doubts about the study's rigor and contribution to the existing literature:

• The introduction discusses AI/ChatGPT adoption, but no adoption theory has been utilized to assess attitudes toward ChatGPT. The absence of a theoretical framework weakens the study's foundation.

• The research gap is not clearly established, making it unclear why this study was conducted. Without a well-defined gap, the study’s objectives seem questionable.

• Numerous studies already exist on the adoption of ChatGPT, particularly regarding knowledge and attitudes. The authors have not adequately demonstrated how this study differs from previous research. Specifically, it is unclear whether findings from nursing students would differ significantly from those of students in other domains.

• Although the study focuses on nursing students, no domain-specific questions were included. This omission limits the study's relevance to nursing education.

• The sample size raises concerns. The manuscript first states that 240 students participated but then mentions that the minimum recommended sample size was 169. If the entire population was invited to participate, why was a sample-based inference needed? Additionally, increasing the sample size solely to enhance statistical significance without considering response rates and data collection timelines is problematic.

• The analysis appears overly simplistic, relying on basic statistical techniques such as ANOVA and elementary graphs. A more robust analytical approach would have strengthened the findings.

Given these concerns, I do not recommend the manuscript for publication in a quality journal.

Recommended Readings:

• Ahmed, F. R., Rushdan, E. E., Al-Yateem, N., Almaazmi, A. N., Subu, M. A., Hijazi, H., ... & Aburuz, M. E. (2024). AI in higher education: unveiling nursing students' perspectives on ChatGPT's challenges and opportunities. Teaching and Learning in Nursing.

• Arjanto, P., & Aditama, M. H. R. (2025). AI in nursing education: From ethical challenges to institutional strategies for effective adoption. Teaching and Learning in Nursing.

• Wang, X., Fei, F., Wei, J., Huang, M., Xiang, F., Tu, J., ... & Gan, J. (2024). Knowledge and attitudes toward artificial intelligence in nursing among various categories of professionals in China: a cross-sectional study. Frontiers in Public Health, 12, 1433252.

• Sallam, M., Salim, N. A., Barakat, M., Al-Mahzoum, K., Ala'a, B., Malaeb, D., ... & Hallit, S. (2023). Assessing health students' attitudes and usage of ChatGPT in Jordan: validation study. JMIR Medical Education, 9(1), e48254.

• Almogren, A. S., Al-Rahmi, W. M., & Dahri, N. A. (2024). Exploring factors influencing the acceptance of ChatGPT in higher education: A smart education perspective. Heliyon, 10(11).

• Sahari, Y., Al-Kadi, A. M. T., & Ali, J. K. M. (2023). A cross sectional study of ChatGPT in translation: Magnitude of use, attitudes, and uncertainties. Journal of Psycholinguistic Research, 52(6), 2937-2954.

6. PLOS authors have the option to publish the peer review history of their article (what does this mean? ). If published, this will include your full peer review and any attached files.

**Do you want your identity to be public for this peer review?** For information about this choice, including consent withdrawal, please see our Privacy Policy .

Reviewer #1: **Yes: ** Moustaq Karim Khan Rony

Reviewer #2: No

---

## [Author Response · Author response to Decision Letter 1]

26 Mar 2025

Dear respected Reviewers

We sincerely appreciate your time and effort in reviewing our manuscript. We are grateful for your insightful comments and constructive feedback, which have significantly improved the clarity, rigor, and impact of our study. We have considered each point raised and have made the necessary revisions accordingly. Below, we provide detailed responses to each comment, indicating how the manuscript has been revised to address the concerns. All modifications done are colored blue.

We believe the revisions have significantly improved the clarity, rigor, and contribution of our work. We respectfully request reconsideration of our manuscript for publication and remain open to any further revisions that may be required.

Thank you again for your guidance and the opportunity to revise and improve our manuscript.

Reviewer #1 Comments

1. The qualitative analysis mentions Braun and Clarke’s framework but does not specify how intercoder reliability was ensured. Were multiple researchers involved in coding? If so, how were disagreements resolved? Providing more details would enhance the trustworthiness of the findings.

Authors’ Response:

Thank you for this suggestion. We have now provided a more detailed explanation of our approach to intercoder reliability in the Methods section. Specifically, two independent researchers manually coded the data. They initially worked separately to identify themes, then compared their findings and discussed discrepancies until reaching consensus. This process ensured consistency and minimized bias, thereby enhancing the trustworthiness of our qualitative analysis.

2. While the study acknowledges certain limitations, it should discuss how these might impact the findings and interpretations. For instance, the sample includes students and educators from a single institution. How might this affect the generalizability of the results? Consider discussing whether similar findings could be expected in other cultural or educational settings.

Authors’ Response:

We appreciate this recommendation. We have expanded the Limitations section to discuss how the single-institution sample may impact generalizability. While the findings provide valuable insights, they may not be fully generalizable to other nursing programs with different curricula, institutional cultures, or technological infrastructures. Future research involving diverse educational settings and international comparisons is recommended to validate and extend our findings.

3. The ethical concerns regarding ChatGPT’s use (plagiarism, over-reliance, and privacy issues) are well-identified. However, the study would benefit from a deeper discussion on potential solutions. Could existing ethical frameworks in nursing education provide guidance?

Authors’ Response:

Thank you for highlighting this point. We have expanded our discussion to incorporate ethical frameworks from nursing education, such as the American Nurses Association (ANA) Code of Ethics and the International Council of Nurses (ICN) Code of Ethics. These frameworks emphasize responsible AI use, professional accountability, and ethical decision-making. Additionally, we have suggested institutional policies, academic integrity guidelines, and structured AI literacy training programs to mitigate ethical risks and ensure responsible ChatGPT use in nursing education.

Reviewer #2 Comments

1. The introduction discusses AI/ChatGPT adoption, but no adoption theory has been utilized to assess attitudes toward ChatGPT. The absence of a theoretical framework weakens the study's foundation.

Authors’ Response:

We appreciate this important observation. In response, we have now explicitly incorporated the Technology Acceptance Model (TAM) as the theoretical framework guiding our study. TAM provides a structured approach to understanding users’ perceptions, attitudes, and adoption of ChatGPT in academic settings. We have revised the Introduction section to reflect this addition.

2. The research gap is not clearly established, making it unclear why this study was conducted. Without a well-defined gap, the study’s objectives seem questionable.

Authors’ Response:

Thank you for this valuable feedback. We have revised the Introduction section to clearly articulate the research gap. Our study uniquely addresses this gap by exploring ChatGPT’s role within the discipline-specific context of nursing education with both students and educators ample and mixed-methods approach to complement and integrate findings.

3. Numerous studies already exist on the adoption of ChatGPT, particularly regarding knowledge and attitudes. The authors have not adequately demonstrated how this study differs from previous research. Specifically, it is unclear whether findings from nursing students would differ significantly from those of students in other domains.

Authors’ Response:

We appreciate this concern. We have now included a comparative discussion in the Introduction and Discussion sections, highlighting how our study differs from prior research. While general AI adoption studies exist, our research focuses on nursing education, a field with unique ethical, practical, and critical thinking demands. The findings provide domain-specific insights that distinguish nursing students' perspectives from those in other academic disciplines.

4. Although the study focuses on nursing students, no domain-specific questions were included. This omission limits the study's relevance to nursing education.

Authors’ Response:

Thank you for this observation. All participants were nursing students and educators, ensuring that responses were inherently discipline-specific. We explicitly use the word "nursing" in our qualitative questions. We have clarified this in the Methods section.

5. The sample size raises concerns. The manuscript first states that 240 students participated but then mentions that the minimum recommended sample size was 169. If the entire population was invited to participate, why was a sample-based inference needed?

Authors’ Response:

We appreciate the reviewer’s observation regarding sample size clarification. While the minimum recommended sample size was calculated to be 169 based on power analysis, we invited the entire student population (N = 300) to participate in order to enhance representativeness and reduce selection bias. A total of 240 students responded, exceeding the minimum requirement and thereby strengthening the reliability and generalizability of the findings.

Sample-based inference was still appropriate because not all members of the population responded. Thus, statistical inference was necessary to estimate the characteristics of the broader student population and examine associations among variables. This approach aligns with best practices in survey-based educational research.

6. The analysis appears overly simplistic, relying on basic statistical techniques such as ANOVA and elementary graphs. A more robust analytical approach would have strengthened the findings.

Authors’ Response:

Thank you for this valuable feedback. In response, we have expanded the statistical analysis to include Pearson correlation and multiple regression analyses in addition to the originally reported ANOVA. These advanced methods allow us to explore predictive relationships and associations between variables more robustly. We have also added five quantitative tables, four figures, and one qualitative table with a graphical summary to enhance data visualization and interpretation. These enhancements provide a deeper and more rigorous analysis of ChatGPT adoption in nursing education and strengthen the overall credibility and impact of our findings.

Authors’ Response to Reviewer 2’s Final Comment:

We appreciate the reviewer’s critical feedback and acknowledge the concerns raised. We have taken substantial steps to improve the manuscript by addressing all comments. While we recognize that the reviewer initially did not recommend the manuscript for publication, we genuinely value constructive criticism, as it allows us to refine and enhance our work. The revisions we have made—including strengthening the statistical analyses, incorporating a theoretical framework, clarifying the research gap, and refining discussions on ethical considerations—aim to elevate the study’s rigor and relevance. Beside the nature of mixed-methods study and its value.

We sincerely hope that these improvements align with the journal's quality expectations. We kindly request reconsideration of our revised manuscript and welcome any further suggestions to enhance its contribution to the field.

We hope that our revisions have sufficiently addressed all concerns. Thank you again for your constructive feedback and guidance. We look forward to your further evaluation.

Sincerely,

Corresponding Author: Prof. Ebtsam

---

## [Decision Letter · Decision Letter 1]

Embracing AI in academia: a mixed methods study of nursing students' and educators' perspectives on using chatgpt

PONE-D-24-60162R1

Dear Dr. Ebstam Aly Abou Hashish

We’re pleased to inform you that your manuscript has been judged scientifically suitable for publication and will be formally accepted for publication once it meets all outstanding technical requirements.

Kind regards,

Mary Mathew, MD

Academic Editor

PLOS ONE

Additional Editor Comments (optional):

Reviewers' comments:

Reviewer's Responses to Questions

**Comments to the Author**

1. If the authors have adequately addressed your comments raised in a previous round of review and you feel that this manuscript is now acceptable for publication, you may indicate that here to bypass the “Comments to the Author” section, enter your conflict of interest statement in the “Confidential to Editor” section, and submit your "Accept" recommendation.

Reviewer #2: All comments have been addressed

2. Is the manuscript technically sound, and do the data support the conclusions?

Reviewer #2: Yes

3. Has the statistical analysis been performed appropriately and rigorously? 

Reviewer #2: Yes

4. Have the authors made all data underlying the findings in their manuscript fully available?

Reviewer #2: Yes

5. Is the manuscript presented in an intelligible fashion and written in standard English?

Reviewer #2: Yes

6. Review Comments to the Author

Reviewer #2: I am pleased with the author’s responses to our queries and the revisions made. The manuscript employs a rigorous methodology and represents a high-quality contribution; accordingly, I recommend its acceptance in its current form.

7. PLOS authors have the option to publish the peer review history of their article (what does this mean? ). If published, this will include your full peer review and any attached files.

**Do you want your identity to be public for this peer review?** For information about this choice, including consent withdrawal, please see our Privacy Policy .

Reviewer #2: **Yes: ** Dr. Puneet Kumar Gupta

---

## [Editor Report · Acceptance letter]

PONE-D-24-60162R1

PLOS ONE

Dear Dr. Abou Hashish,

I'm pleased to inform you that your manuscript has been deemed suitable for publication in PLOS ONE. Congratulations! Your manuscript is now being handed over to our production team.

Kind regards,

on behalf of

Dr. Mary Mathew

Academic Editor

PLOS ONE
